# Human kinetochores are swivel joints that mediate microtubule attachments

Chris A Smith[1,2], Andrew D McAinsh[1]*, Nigel J Burroughs[3,4]*

[1]Centre for Mechanochemical Cell Biology, Division of Biomedical Sciences, Warwick Medical School, University of Warwick, Coventry, United Kingdom; [2]Molecular Organisation and Assembly in Cells (MOAC) Doctoral Training Centre, University of Warwick, Coventry, United Kingdom; [3]Warwick Systems Biology Centre, University of Warwick, Coventry, United Kingdom; [4]Mathematics Institute, University of Warwick, Coventry, United Kingdom

**Abstract** Chromosome segregation is a mechanical process that requires assembly of the mitotic spindle – a dynamic microtubule-based force-generating machine. Connections to this spindle are mediated by sister kinetochore pairs, that form dynamic end-on attachments to microtubules emanating from opposite spindle poles. This bi-orientation generates forces that have been reported to stretch the kinetochore itself, which has been suggested to stabilise attachment and silence the spindle checkpoint. We reveal using three dimensional tracking that the outer kinetochore domain can swivel around the inner kinetochore/centromere, which results in large reductions in intra-kinetochore distance (delta) when viewed in lower dimensions. We show that swivel provides a mechanical flexibility that enables kinetochores at the periphery of the spindle to engage microtubules. Swivel reduces as cells approach anaphase, suggesting an organisational change linked to checkpoint satisfaction and/or obligatory changes in kinetochore mechanochemistry may occur before dissolution of sister chromatid cohesion.

*For correspondence: A.D. McAinsh@warwick.ac.uk (ADM); N.J.Burroughs@warwick.ac.uk (NJB)

**Competing interests:** The authors declare that no competing interests exist.

## Introduction

Segregating multiple chromosomes into daughter cells is a major engineering challenge for the human cell: replicated chromosomes (sister chromatids) must form physical attachments to dynamic microtubules, which are organised into a bipolar scaffold (*Dumont and Mitchison, 2009*). Kinetochores mediate this attachment, and harness the pushing and pulling forces associated with microtubule polymerisation/depolymerisation to power chromosome movement (*Rago and Cheeseman, 2013*). Extensive biochemical work has assembled a detailed parts list for the kinetochore and we now understand that the core structure assembles in a hierarchical fashion with the CCAN (constitutive centromere associated network) physically bridging CENP-A containing chromatin and the microtubule-binding KMN network (KNL1, MIS12 and NDC80 complexes) (*Pesenti et al., 2016*). Multiple copies of each component (*Suzuki et al., 2015*) produce a supramolecular disk-shaped assembly with a (metaphase) diameter of ~200 nm and a depth of ~100 nm (*Rieder and Salmon, 1998*).

Insight into the *in vivo* mechanical properties of the kinetochore have come from elegant experiments in which two different components of the kinetochore along the chromosome-to-microtubule binding axis are labelled with different fluorophores, each producing a diffraction-limited spot. The distance between the spot centres is the "*intra-kinetochore distance* (delta; Δ)" (*Figure 1A*). Removing microtubule-pulling forces in both human and Drosophila cells has been shown to reduce the distance between centromeric chromatin and the microtubule-binding outer layer of the kinetochore by ~30 nm (*Wan et al., 2009*; *Maresca and Salmon, 2009*). This is interpreted as evidence of

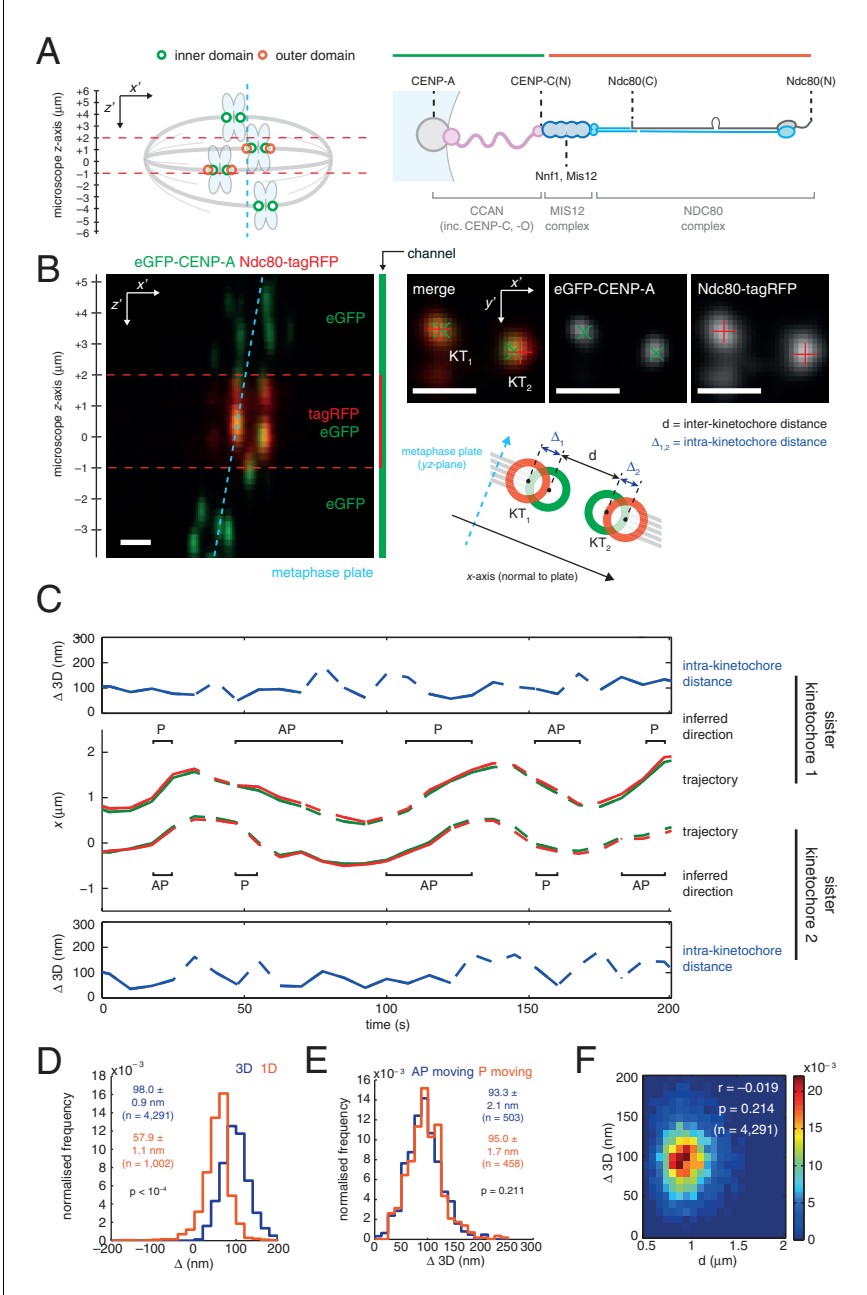

**Figure 1.** 3D dual colour kinetochore tracking assay demonstrates rigid intra-kinetochore structure. (**A**) Schematic of imaging setup for tracking of fluorescently marked inner kinetochore (green circles) and outer kinetochore domains (red circles) within a 12 μm z-stack to measure intra-kinetochore distance, Δ, in HeLa-K cells. Schematic showing approximate architecture of the mammalian kinetochore including CENP-A nucleosomes (grey circle), CCAN (pink), MIS12 complex (dark blue) and the NDC80 complex (light blue with the Ndc80 subunit highlighted in grey). (**B**) Live cell imaging of eGFP-CENP-A and Ndc80-tagRFP in microscope x'z' plane demonstrates imaging of tagRFP in the central 3 μm of the z'-stack. Images in microscope x'y' plane demonstrate Gaussian-fitted spot centres. Schematic shows metaphase plate coordinate system, [x,y,z] (x is normal to the metaphase plate, y is the line intersection of the metaphase plate and the x'y' plane, and z is orthogonal to both to make a right-handed coordinate system), and measurements of intra- (Δ) and inter- (d) kinetochore distances. Scale bars 500 nm. (**C**) Trajectory of a kinetochore pair's eGFP-CENP-A (green lines) and Ndc80-tagRFP signals (red lines) in x, with measurements of Δ 3D for each sister, where dashed lines represent time points removed in quality control (see Materials and methods). Regions of poleward (P) and away-from-the-pole (AP) movement are labelled. (**D**) Distribution of measurements of Δ in 3D (blue; n = 4291), and 1D measurement of Δ (gold; n = 1002) used in

*Figure 1 continued*

previous studies. (**E**) Distribution of measurements of Δ 3D in AP- (blue; n = 503) and P-moving kinetochores (gold; n = 458), demonstrating no significant difference (p = 0.211). Values given in **D** and **E** are medians ± standard error. (**F**) 2D histogram of Δ 3D against *d*, demonstrating no correlation (r = –0.019, p = 0.214; n = 4291).

The following source data and figure supplements are available for figure 1:

**Source data 1.** Intra-kinetochore distances with and without microtubule attachment.

**Figure supplement 1.** Inter- and intra-kinetochore distances during directed chromosome motion.

**Figure supplement 2.** Intra-kinetochore distances with and without microtubule attachment.

**Figure supplement 3.** The experimental setup to measure intra-kinetochore distance from single time point *z*-stacks.

**Figure supplement 4.** Chromatic shift correction precision and accuracy in the dynamic live assay to analyse intra-kinetochore measurements.

tension within the kinetochore generated by microtubule pulling, an event that has been linked to how kinetochores stabilise correct microtubule attachments and silence the spindle assembly checkpoint (*Joglekar et al., 2009*; *Maresca and Salmon, 2009*; *Uchida et al., 2009*; *Drpic et al., 2015*). However, the requirement for this intra-kinetochore tension in the checkpoint silencing mechanism has recently been challenged (*Etemad et al., 2015*; *Tauchman et al., 2015*; *Magidson et al., 2016*).

Counter intuitively, live cell imaging of proteins in the inner and outer domains of the kinetochore in PtK2 cells revealed that the kinetochore is compressed (rather than stretched) when under pulling force (*Dumont et al., 2012*). Moreover, following bi-orientation Δ does not increase or decrease as the inter-sister distance increases (the latter is known to vary under the forces acting on kinetochores during metaphase oscillations) suggesting that the kinetochore is now non-compliant (*Suzuki et al., 2014*). One possible explanation for these differences is the reliance on 1D or 2D measurement of the 3D movement of kinetochores within the spindle, while fixation conditions and the tilting of the kinetochore have also been found to have effects on Δ measurements (*Magidson et al., 2016*; *Wan et al., 2009*). Thus, live cell monitoring of $\Delta_{3D}$ will be an important next step to understanding the micro-mechanics of the kinetochore.

## Results and discussion

To measure Δ in 3D we used eGFP-CENP-A (named CENP-A from here on) to mark the inner kinetochore domain and the carboxy (C)-terminus of the Ndc80 protein with tagRFP (Ndc80(C)) to mark the outer domain (*Figure 1A and B*) and tracked the spots in 3D with sub-pixel accuracy using a two-camera setup on a spinning disk confocal microscope (*Figure 1C*; see Materials and methods for details, including 3D chromatic aberration correction). We confirmed that kinetochore dynamics were normal following exogenous expression of Ndc80-tagRFP (*Figure 1—figure supplement 1A and B*). The 3D Euclidean distance Δ between eGFP and tagRFP spot centres ($\Delta_{3D}$) for sister kinetochore pairs in live cells was estimated to be 98.0 (± 0.9) nm (n = 4291; *Figure 1D*). Unlike in PtK2 cells (*Dumont et al., 2012*) we did not detect a difference between away-from-the-pole (AP) and poleward (P) moving kinetochores in either 3D (*Figure 1E*) or 2D measurements (*Figure 1—figure supplement 1C*). Our 3D measurement is somewhat larger than the ~60 nm predicted from previous measurements of CENP-A-to-Spc24 distance (*Suzuki et al., 2014*) and the position of the Ndc80 C-terminus based on electron microscopy of the purified complex (*Wei et al., 2005*). However, the CENP-A-to-Spc24 measurement in Suzuki *et al.* was effectively one-dimensional ($\Delta_{1D}$, see supplemental discussion); applying the same 1D method to our data gives a distance of 57.9 (± 1.1) nm (n = 1002; *Figure 1D*) indicating that projection effects (3D to 1D) may be giving a misleading picture of kinetochore structure/architecture.

To examine compliance under tension we analysed the correlation of $\Delta_{3D}$ with the changes in inter-kinetochore distance that take place during kinetochore oscillations in metaphase. We found no correlation (*Figure 1F*) indicating that the CENP-A-to-Ndc80(C) connection is indeed non-compliant (stiff). To test this under more extreme changes of applied load, we tested how loss of microtubule attachment would affect $\Delta_{3D}$ by treating cells with 3 µM nocodazole for 2 hr (*Figure 2A*, and *Figure 2—figure supplement 1*). We found that despite a 30% decrease (283 nm) in inter-kinetochore distance, *d* (*Figure 2B*), $\Delta_{3D}$ was only marginally reduced by 5% (5 nm, significant at p = 0.012, n ≥ 649 for each condition). This small change in $\Delta_{3D}$ for the CENP-A-to-Ndc80(C) linkage is consistent with the only other live measurement of $\Delta$ in human cells (between CENP-A and mCherry-Mis12) where $\Delta_{2D}$ decreased marginally under nocodazole (*Uchida et al., 2009*). Our own 3D measurement between these same markers, and also between GFP-CENP-C and Ndc80(C), produced the same marginal 5–7 nm reductions in $\Delta_{3D}$ (*Figure 1—figure supplement 2*, and *Figure 1—source data 1*). In paraformaldehyde fixed cells the distance between CENP-A and the position of the MIS12 complex (using anti-Nnf1 antibodies) was also reduced by 5 nm, while both eGFP-CENP-A and endogenous CENP-A (using anti-CENP-A antibodies) to Ndc80(C) give marginal changes (*Figure 1—figure supplement 2* and *Figure 1—source data 1*).

Interestingly, the CENP-A-to-Ndc80(C) linkage was not affected following treatment with 10 µM taxol (p = 0.302; *Figure 2—figure supplement 2*). The spindle is retained under these conditions although microtubule dynamics are inhibited – reflected by the decrease in sister separation (*Figure 2—figure supplement 2*). This demonstrates that exerting pulling force on the kinetochore is not sufficient to change $\Delta_{3D}$ and suggests that it is the presence of microtubules that induces a conformational change in the kinetochore.

We next tested whether there are changes in the outer domain of the kinetochore by determining the distance from eGFP-CENP-A and GFP-CENP-C to the amino (N)-terminal end of Ndc80 in live cells. We detected a ~15 nm reduction in $\Delta_{3D}$ following nocodazole treatment, with up to half of this contributed from the change in the inner domain discussed above (*Figure 1—figure supplement 2*, and *Figure 1—source data 1*). The additional movement of the Ndc80 N-terminus may relate to the conformational change in the NDC80 complex that has been reported in budding yeast (*Aravamudhan et al., 2015*) although further experiments will be needed to confirm this. This delta change is smaller than that seen in fixed cells (see *Etemad et al., 2015*, and *Figure 1—source data 1*) a difference that may reflect the reported effects of fixation on kinetochore integrity/epitope accessibility (*Magidson et al., 2016*). We conclude that microtubule-pulling forces do not dramatically stretch the kinetochore and that the intra-kinetochore linkages are relatively stiff. This is consistent with previous work showing that the outer domain is non-compliant (*Suzuki et al., 2014*).

These changes in $\Delta_{3D}$ are in stark contrast to the large changes we observe in projected distances, $\Delta_{1D}$ and $\Delta_{2D}$, under nocodazole treatment (*Figure 2—source data 1*). For example, the CENP-A-to-Ndc80(C) measurement for $\Delta_{1D}$ decreased by 70% from 57.9 (± 1.1) nm in untreated cells to 17.8 (± 4.4) nm in nocodazole treated cells. This 40 nm change observed in 1D is considerably more than the 5 nm (5%) changes in $\Delta_{3D}$. The observed discrepancy between $\Delta_{1D}$ and $\Delta_{3D}$ suggests that a rotation is present and that this is the major degree of freedom upon microtubule attachment (rather than the stretching of the kinetochore). This rotation of the outer kinetochore around the inner kinetochore is apparent in images of the eGFP-CENP-A and Ndc80-tagRFP signals, which are rarely perfectly co-linear (*Figure 2D*). This is not due to the kinetochore markers used because the same effect is seen when observing other inner/outer kinetochore markers (*Figure 2—figure supplement 3*). Thus, $\Delta_{1D}$ underestimates the distance between the inner and outer kinetochore because the outer kinetochore exhibits substantial rotation (*Figure 2D*).

We sought to investigate this new degree of freedom, a phenomenon we call swivel. Swivel was calculated as the angle tended by the vector between the paired sisters' eGFP signals (sister-sister axis) and the vector between the eGFP and tagRFP signals on each kinetochore (the kinetochore axis, defined by the inner/outer domain marker axis) (*Figure 3A*). This degree of freedom is distinct from the twist angle of a kinetochore pair, *i.e.* the angle tended between the sister-sister axis and the normal to the metaphase plate. Calculation of swivel across all untreated kinetochores yielded a median of 51.6° as compared to 73.8° in 3 µM nocodazole (*Figure 3B*) – this difference is significant (p < 10⁻⁴) and explains the decrease in the 1D measurement of $\Delta$ in nocodazole-treated cells. To remove effects due to the lower resolution in the z-axis, swivel was decomposed into components; we defined the *y*-directional swivel as the projection onto the *xy*-plane. This measurement showed

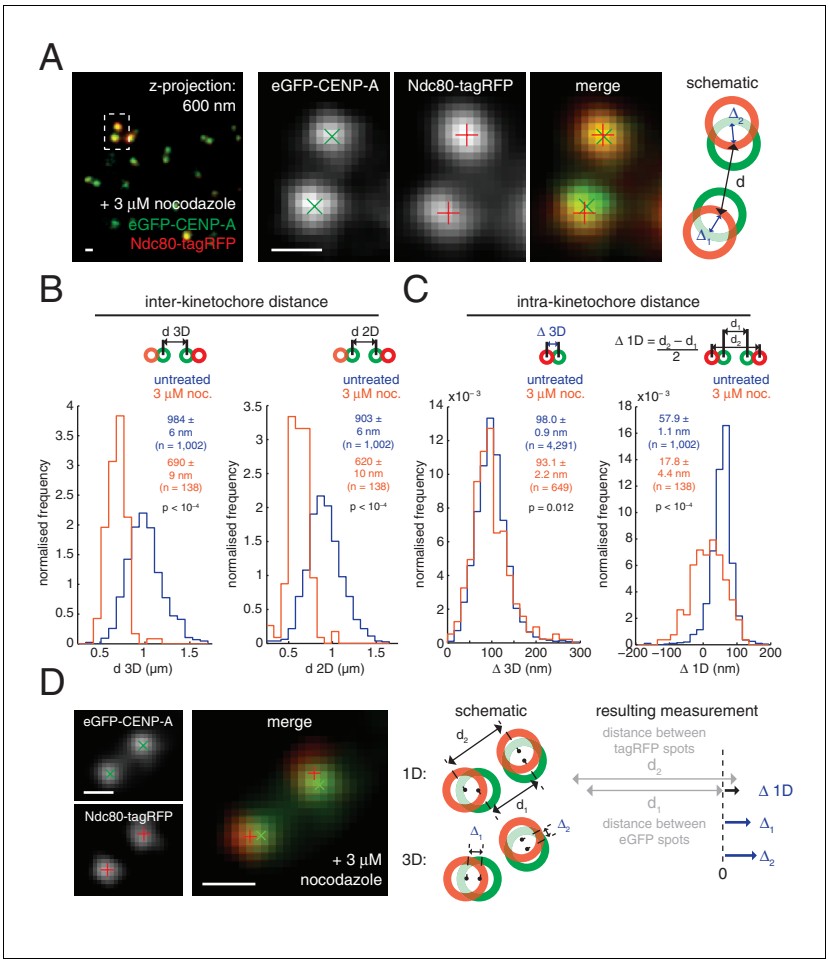

**Figure 2.** Nocodazole treatment marginally decreases 3D intra-kinetochore distance. (**A**) Live cell imaging of eGFP-CENP-A and Ndc80-tagRFP after 3 µM nocodazole treatment (600 nm z-projection). Representative images of a kinetochore pair demonstrating Gaussian-fitted spot centres. Schematic of the kinetochore pair shows measurements of intra-, Δ, and inter-kinetochore distances, d. Scale bars 500 nm. (**B**) Distribution of measurements of inter-sister distance d in 3D (left panel) and 2D (right panel), in untreated (blue; n = 1002) and 3 µM nocodazole treated cells (gold; n = 138); both are significantly different (p < 10⁻⁴). (**C**) Distribution of measurements of intra-kinetochore distance Δ in 3D (left panel) and 1D (right panel), in untreated (blue; n₃D = 4291, n₁D = 1002) and 3 µM nocodazole-treated cells (gold; n₃D = 649, n₁D = 138); both are significantly different to 95% confidence. In (**B**) and (**C**), schematics illustrate measurement method, statistical tests were Mann-Whitney U tests, and values given are medians ± standard error. (**D**) Example of a kinetochore pair demonstrating rotation of the kinetochore axis relative to the sister-sister axis. Schematic illustrates how 1D and 3D measurements of Δ are made, and the resulting underestimate of Δ 1D. Scale bars 500 nm.

The following source data and figure supplements are available for figure 2:

**Source data 1.** Swivel increases under nocodazole treatment for multiple kinetochore markers.

**Figure supplement 1.** Quantification of spindle depolymerisation following nocodazole treatment.

**Figure supplement 2.** Effect of Taxol on inter- and intra-kinetochore distance and swivel.

**Figure supplement 3.** Kinetochore swivel is also present between other pairs of kinetochore markers.

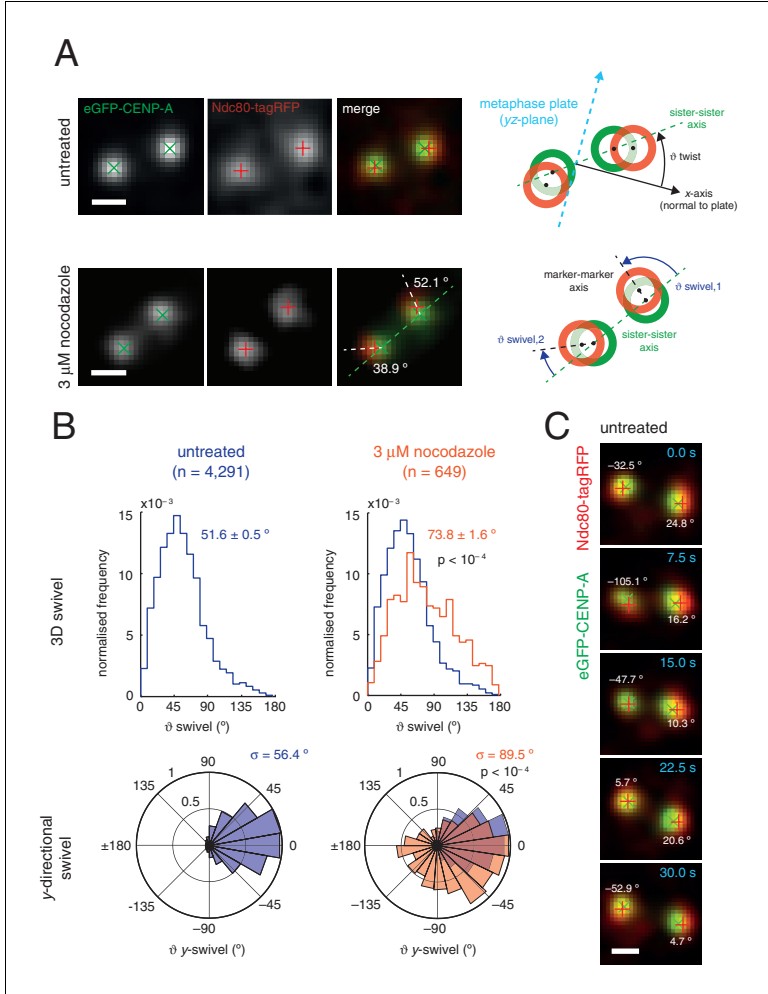

**Figure 3.** Outer kinetochore components are capable of 'swivel' about the inner kinetochore, which increases upon microtubule depolymerisation. (**A**) Example images of kinetochore pairs in untreated (top) and 3 µM nocodazole (bottom) cells exhibiting swivel in each of its kinetochores. Green and white dashed lines represent sister-sister and intra-kinetochore axes, respectively. Associated schematic shows measurements of swivel ($\vartheta$ swivel, between the sister-sister and marker-marker axes), and twist ($\vartheta$ twist, between the sister-sister axis and the metaphase plate). Stated values given are $\vartheta y$-swivel at each kinetochore. Scale bars 500 nm. (**B**) Distributions of $\vartheta$ swivel in 3D (histograms in top row) and projected on the $y$-axis (rose plots in bottom row), in untreated (blue; n = 4291) cells compared to 3 µM nocodazole (gold; n = 649), each significantly different to untreated (p < $10^{-4}$). Values given are medians ± standard error in $\vartheta$ swivel, and standard deviation, $\sigma$, for $\vartheta y$-swivel. Statistical tests were Mann-Whitney U for $\vartheta$ swivel, and F test for $\vartheta y$-swivel. (**C**) A kinetochore pair in untreated cells exhibiting temporally-changing swivel. Time is given at the top right of each frame. Values given are $\vartheta y$-swivel at each kinetochore for each time point. Scale bar 500 nm.

The following figure supplement is available for figure 3:

**Figure supplement 1.** Measurement of axial swivel, and characterisation of 1D delta's underestimate of 3D delta.

that 3 µM nocodazole treatment results in a larger range of $y$-directional swivel (population standard deviation = 56.4° in untreated vs. 89.5° in 3 µM nocodazole), with some outer kinetochores capable of rotating inside the eGFP signal (i.e. $y$-swivel > 90°; *Figure 3C*), which explains the presence of negative measurements for $\Delta_{1D}$. We confirmed the increase in swivel between GFP-CENP-C and Ndc80(C) in live cells and between CENP-A and the MIS12 complex in paraformaldehyde-fixed cells treated with nocodazole (*Figure 2—source data 1*). This increase in 3D swivel and broadening of $y$-directional swivel using CENP-A-to-Ndc80(C) were also seen in 10 µM taxol-treated cells, albeit to a

lesser extent (*Figure 2—figure supplement 2*). This condition leaves the spindle intact and suggests that loss of microtubule pulling force is sufficient to increase swivel, although loss of microtubule-kinetochore attachment increases swivel still further. Furthermore, *y*-swivel changes over time demonstrating that kinetochores do not have a fixed swivel, swivel potentially adapting to changes in microtubule attachment, microtubule forces and sister kinetochore dynamics (*Figure 3C*).

To ascertain the role of swivel, we observe that the kinetochore-to-pole angle decreases in the spindle towards the equatorial plane periphery, being collinear at the centre; swivel may thus enable kinetochores to correctly engage k-fibres across the metaphase plate (see *Figure 1A*). We found that kinetochore pairs at the centre of the metaphase plate exhibited a smaller degree of swivel compared to those at the periphery (*Figure 4A and B*). Plotting *y*-directional swivel against the position along the metaphase plate in the *y*-axis demonstrated that kinetochores begin to exhibit swivel towards their attached spindle pole (*Figure 4B*), a phenomenon that is more evident once kinetochore pairs are beyond 4 µm from the centre of the metaphase plate. To confirm this we performed immunofluorescence to visualise the k-fibres using anti-tubulin antibodies under paraformaldehyde fixation (*Figure 2—figure supplement 1A*). The eGFP, tagRFP and k-fibre signals were approximately co-linear, with the angles tended on the inter-sister axis by the k-fibre ($\vartheta_{kMT}$) and intra-kinetochore axes ($\vartheta_{y\text{-swivel}}$) being significantly correlated ($r = 0.371$, $p < 10^{-4}$; *Figure 4C*, and *Figure 4—figure supplement 1*). This shows that outer kinetochore swivel can be used as a proxy for the direction in which the k-fibres are bound by the kinetochore – this is not surprising as Ndc80 binds microtubules directly.

Large changes in intra-kinetochore stretch have also been reported to be associated with the mechanism that satisfies the spindle assembly checkpoint, SAC (*Maresca and Salmon, 2009*; *Uchida et al., 2009*). Our live cell data questions this model and thus prompted us to investigate whether outer domain swivel was correlated with anaphase onset. To do this we took advantage of the observation that the thickness of the metaphase plate reduces as a cell transits metaphase and approaches anaphase onset (*Jaqaman et al., 2010*). We plotted the *y*-directional swivel against the position of a kinetochore in the *y*-direction and the thickness of the metaphase plate. Fitting a sheet to this data reveals how swivel clearly decreases as a cell moves through metaphase (*Figure 4D*). Our previous observation that swivel increases towards the periphery of the plate holds true across the range of plate thicknesses, although the range over which swivel increased at the plate periphery decreased with decreasing plate thickness (*Figure 4D*). Binning the plate thicknesses allowed a statistical comparison that peripheral kinetochores decreased absolute *y*-swivel from 45.7 (± 1.9) ° (n = 455) in early metaphase to 33.6 (± 1.4) ° (n = 460) in late metaphase ($p < 10^{-4}$; *Figure 4E and F*, and *Figure 4—source data 1*). We can also confirm that $\Delta_{1D}$ increases from early to late metaphase (*Figure 4E*, and *Figure 4—source data 1*), again suggesting that swivel can explain changes in intra-kinetochore stretch (1D) that have been linked to SAC silencing. On the other hand, $\Delta_{3D}$ (CENP-A-to-Ndc80(C)) does not increase as metaphase progresses (*Figure 4E*), further highlighting how swivel is the distinctive mechanical change in kinetochore structure that correlates with time of anaphase onset.

At this stage it is unknown whether the swivel of the outer kinetochore has any direct role in the SAC silencing mechanism or whether it is a downstream event associated with the falling Cdk1-CyclinB activity that follows SAC satisfaction (*Clute and Pines, 1999*). What our data does suggest is that changes in swivel, being substantially greater than 3D distance changes, provide a proximity change between Aurora B and its substrates. How swivel relates to aurora kinase activity during early mitosis when error correction is active will be an important area for future study. It is also tempting to speculate that kinetochores are undergoing a shift in their mechanochemical status in readiness for the imminent onset of anaphase. As well as the swivel changes reported here, this may also directly relate to the observation that kinetochore speeds decrease as anaphase approaches (*Jaqaman et al., 2010*).

From a mechanical perspective, the swivelling of the outer kinetochore requires a flexible structure, that at the same time is non-compliant under varying load. One possibility is that the outer kinetochore can move and remodel relative to the fixed surface of the inner kinetochore (*Figure 4G (ii)*) as recently shown (*Wynne and Funabiki, 2015*; *Magidson et al., 2016*). We cannot rule out that we are observing a complete rotation of the whole kinetochore structure within the chromosome (*Figure 4G(ii)*). This idea is supported by the positive correlation in the tilt of elliptical CENP-A and Ndc80 fluorescence signals in fixed cell imaging (*Wan et al., 2009*). The fact that swivelling is most

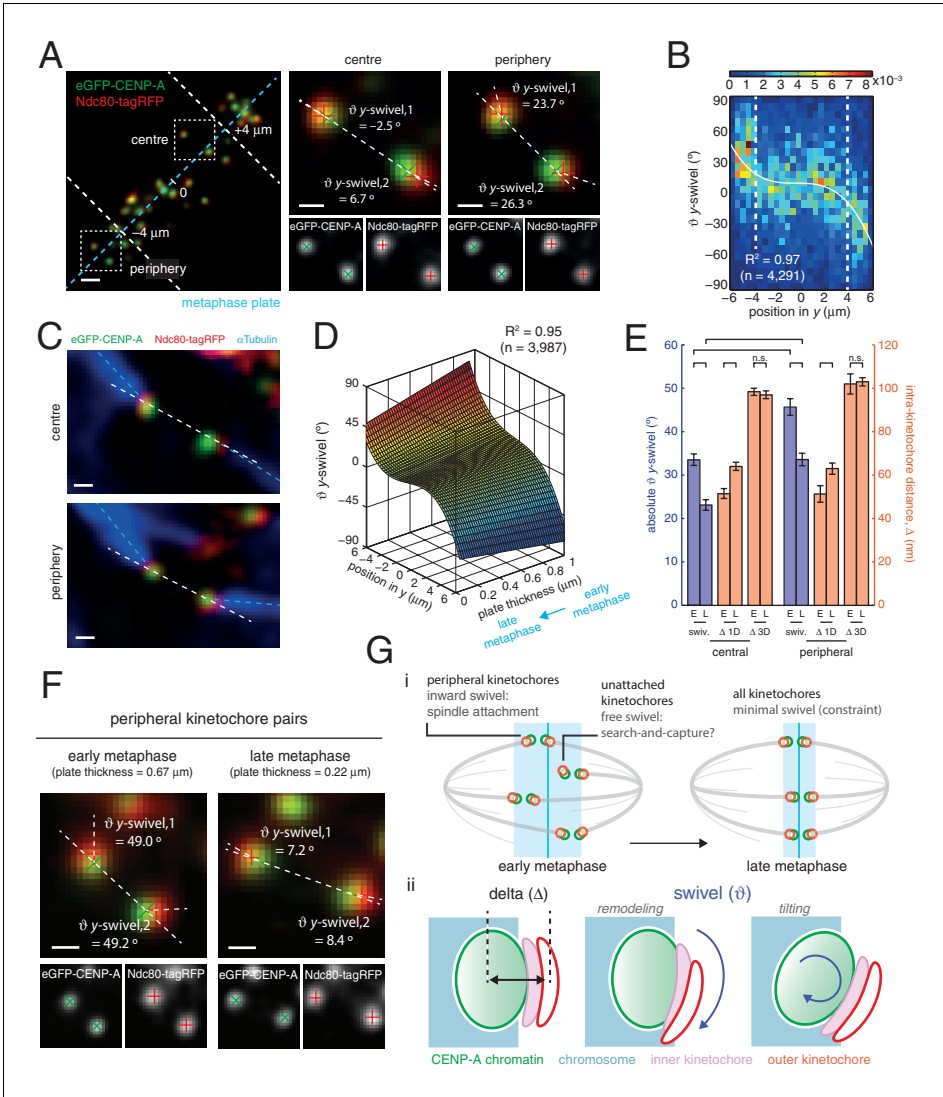

**Figure 4.** Spatial and temporal control of kinetochore swivel. (**A**) Kinetochore pairs in centre (centre) and periphery (right) from a metaphase plate (left) showing Gaussian-fitted spot centres, $\vartheta y$-swivel, sister-sister axis and marker-marker axis. (**B**) 2D histogram of $\vartheta y$-swivel vs. spindle position in $y$ (n = 4291). White dashed lines divide spindle into the central and peripheral regions, separated at 4 μm. White curve is best-fit degree-3 polynomial, with coefficient of determination, $R^2 = 0.97$. (**C**) Pairs of sister kinetochores expressing eGFP-CENP-A, Ndc80-tagRFP and stained with αTubulin antibodies. (**D**) Best-fit sheet through $\vartheta y$-swivel against spindle position in $y$ (degree-3 polynomial) and metaphase plate thickness (degree-1 polynomial; n = 3987), with coefficient of determination, $R^2 = 0.95$. (**E**) Absolute $\vartheta y$-swivel (blue, left axis), and Δ 1D and Δ 3D (gold, right axis), for central and peripheral kinetochores for early (> 0.65) and late (< 0.45) metaphase (E, L, respectively) as determined by plate thickness. Statistical tests were significant (p < $10^{-4}$) except where n.s. (no significance) is stated. Values are given in *Figure 4—source data 1*. (**F**) Examples of kinetochore pairs at spindle periphery during early (left) and late (right) metaphase. (**G**) (**i**) Schematic of dependency of swivel on spatial position, attachment status, and progression through metaphase towards anaphase onset. Light blue box indicates thickness of the metaphase plate. (**ii**) Mechanical degrees of freedom in kinetochore include changes in intra-kinetochore distance (delta; Δ – black arrow) and swivelling ($\vartheta$ – blue arrows), which may involve remodelling of the kinetochore domains and/or tilting of the entire kinetochore structure within the chromosome. All scale bars 500 nm.

The following source data and figure supplement are available for figure 4:

**Source data 1.** Swivel decrease is partnered by an increase in 1D at anaphase onset, but 3D is invariant.

**Figure supplement 1.** Inner and outer kinetochore markers are generally co-linear with their attached k-fibre.

extreme on unattached kinetochores suggests that it may optimise the microtubule search-and-capture process (*Figure 4G(i)*) – this would have a similar effect to broadening of the microtubule-binding region of the kinetochore, as has been implicated in the timing and accuracy of search-and-capture (*Zaytsev et al., 2015*).

Finally, our work now opens up the possibility to map the 3D Euclidean distances between all kinetochore components, and between different domains within proteins and complexes. However, changes in delta under different physiological conditions need to be interpreted with caution as the shape and size of kinetochore domains are subject to change (see *Magidson et al., 2016*). Moreover, the NDC80 complex can be recruited via two distinct physical linkages (CENP-T and CENP-C/Mis12) which may have different compliances/conformational dynamics and further complicate interpretation of delta measurements. Nevertheless, delta and swivel do provide information on the supramolecular organisation of the kinetochore and as such should provide important insights into the mechanical and organisational changes that are taking place during the cycles of microtubule attachment and potential regulatory processes of error-correction and SAC activation/silencing.

## Materials and methods

### Molecular biology

An mCherry-Ndc80 plasmid (pMC299) was constructed by sub-cloning Ndc80 cDNA from pEGFP-Hec1 (kind gift of Eric Nigg) using *KpnI* and *HindIII*. Ndc80 cDNA was then amplified and ligated into tagRFP-N1 (pMC387; pEGFP-N1 (Clontech, Mountain View, CA) in which eGFP was replaced by tagRFP), creating Ndc80-tagRFP (pMC389). A Bub3-eGFP plasmid (pMC360) was generated by amplification by PCR of the short isoform of Bub3 cDNA using forward primer 5'-GCGCTCGAGATGACCGGTTCTAAC-3' and reverse primer 5'-CGCCTGCAGAGTACATGGTGACTT-3', and ligated into pEGFP-N1 (Clontech) using *XhoI* and *PstI*. To generate a HaloTag-CENP-A plasmid (pMC442), full-length CENP-A cDNA (GeneArt #2015ABQWHP, Thermofisher, Waltham, MA) was inserted into a pHTN HaloTag CMV-neo plasmid (Promega, Madison, WI) using *EcoRI* and *XbaI* sites. All constructs were confirmed by sequencing (Source BioScience, UK).

### Cell culture and cell line production

All HeLa-Kyoto (HeLa-K; ATCC® CCL-2™, UK) cells were grown in DMEM media (Fisher, UK) supplemented with 10% foetal calf serum (Fisher), 100 µg ml$^{-1}$ penicillin and 100 µg ml$^{-1}$ streptomycin maintained in 5% $CO_2$ in a humidified incubator at 37°C. eGFP-CENP-A cells (*Jaqaman et al., 2010*) were maintained in 0.1 µg ml$^{-1}$ puromycin (Fisher). mCherry-CENP-A cells (*Armond et al., 2015*) were maintained in 0.1 µg ml$^{-1}$ puromycin (Fisher). eGFP-CENP-A / mCherry-CENP-A cells were maintained in 0.3 µg ml$^{-1}$ puromycin and 300 µg ml$^{-1}$ geneticin. eGFP-CENP-A / mCherry-Mis12 cells (*Uchida et al., 2009*) were maintained in 0.25 µg ml$^{-1}$ puromycin and 500 µg ml$^{-1}$ geneticin. HaloTag-CENP-A cells were maintained in 300 µg ml$^{-1}$ geneticin. GFP-CENP-C cells (*Klare et al., 2015*), GFP-CENP-O cells (*Amaro et al., 2010*) and plain HeLa-K cells were maintained in antibiotic-free medium. The eGFP-CENP-A / mCherry-CENP-A cell line was established by stable transfection of eGFP-CENP-A plasmid into cells stably expressing mCherry-CENP-A. The HaloTag-CENP-A cell line was established by stable transfection of HaloTag-CENP-A plasmid into plain HeLa-K cells. All stable transfections were performed using FuGENE 6 (Promega) and single clones were selected by 300 µg ml$^{-1}$ geneticin. eGFP-CENP-A / mCherry-CENP-A cells were imaged overnight to ensure that mitotic timings were unaffected by the additional incorporation of the fluorescently-tagged CENP-A. For live cell experiments all cell lines were cultured in FluoroDish tissue culture dishes with cover glass bottoms (WPI, Sarasota, FL).

### Transient transfection and drug treatments

Transient transfection of Ndc80-tagRFP was performed by incubating 0.67 µg ml$^{-1}$ Ndc80-tagRFP plasmid and 4.5 µl ml$^{-1}$ FuGENE 6 (Promega) separately in OptiMEM for 5 min, then combined for 30 min before application to cells; antibiotic-free DMEM was transferred to cells prior to addition of transfection reagent to cultured eGFP-CENP-A cells. Transfection was performed for 24 hr prior to drug treatment, or prior to imaging for untreated experiments. Transient transfections of mCherry-Ndc80 and Bub3-eGFP were performed as above, but instead using 3 µl ml$^{-1}$ FuGENE 6 (Promega),

and either 0.33 µg ml⁻¹ Bub3-eGFP plasmid or 0.50 µg ml⁻¹ mCherry-CENP-A plasmid. Our existing kinetochore tracking assay (*Burroughs et al., 2015*) was used to confirm that sister separation remained at ~950 nm and half-period of kinetochore oscillations remained at ~35 s for varying levels of Ndc80-tagRFP expression (*Figure 1—figure supplement 1A and B*), indicating that tagging the C-terminus of Ndc80 did not perturb kinetochore dynamics. This contrasts to previous work showing that attaching fluorescent markers to Ndc80's N-terminus causes perturbed force generation in live cells (*Mattiuzzo et al., 2011*); we only use the mCherry-Ndc80 construct here for completeness. Nocodazole treatments were performed with 3 µM nocodazole (Fisher) for 2 hr prior to imaging. Taxol treatment was performed with 10 µM taxol (Tocris Bioscience, UK) for 1 hr prior to imaging. The loss of microtubules under 2 hr incubation of nocodazole was confirmed by immunofluorescence experiments (*Figure 2—figure supplement 1*).

## Live-cell imaging

All live images and movies were acquired using a confocal spinning-disk microscope (VOX UltraView; PerkinElmer, UK) with a 100X 1.4 NA oil objective and two Hamamatsu ORCA-R2 cameras, controlled by Volocity 6.0 (PerkinElmer) running on a Windows 7 64-bit (Microsoft, Redmond, WA) PC (IBM, New Castle, NY). Camera pixels had an effective pixel size of 69.4 nm in the axial direction, as measured using a microscopic ruler (Pyser SGI, UK). The two cameras were preliminarily aligned using Volocity by aligning 500 nm TetraSpeck fluorescence microspheres (Thermofisher) in the 488 and 561 nm wavelengths. For live cell imaging, a common light path containing two wavelengths was split by a dichroic to each of the two cameras to allow simultaneous imaging of two wavelengths. Cells were first identified using bright-field illumination to minimise phototoxicity. Interphase cells imaged for chromatic shift calculation were imaged over 121 z-slices separated by 100 nm for a single image stack. Laser power in the 488 and 561 nm wavelengths were both set to 15% with exposure time 50 ms per z-slice. Metaphase cells expressing eGFP-CENP-A and Ndc80-tagRFP were imaged over time for dual colour tracking (dynamic live movies), whilst all other metaphase cells were imaged at a single time point for dual colour spot finding (static live images), all utilising the two camera setup. For dynamic live movies, cells were imaged over 61 z-slices separated by 200 nm, for 5 min every 7.5 s (41 time points). Laser power in the 488 nm wavelength was set to 5%. Laser power in the 561 nm wavelength was set to 20% over z-slices 26–41 (from the bottom of the cell), and set to 0% for all other z-slices to minimise phototoxicity. Exposure time was 50 ms per z-slice. For static live images, cells were imaged over 61 z-slices separated by 200 nm for a single time point. Exposure time was 50 ms per z-slice, with laser power in the 488 and 561 nm wavelengths set to: 10% and 30% for eGFP-CENP-A / mCherry-Mis12; 20% and 30% for mCherry-CENP-A / Bub3-eGFP; 30% and 30% for GFP-CENP-C / Ndc80-tagRFP; 30% and 30% for GFP-CENP-C / mCherry-Ndc80; 40% and 20% for GFP-CENP-O / Ndc80-tagRFP, respectively, to optimise imaging. Spot centres were located as in the chromatic shift calculation, described below.

## Immunofluorescence

HeLa-K cells were prepared as above, except glass coverslips were used. Cells underwent pre-extraction for 30 s in fixation buffer (10 mM EGTA, 0.2% Triton-X100, 1 mM MgCl$_2$, and 20 mM PIPES pH 6.8; all Sigma-Aldrich, St. Louis, MO), before addition of 4% paraformaldehyde (Sigma-Aldrich) for 10 min. Cells were then blocked for 30 min in PBS supplemented with 3% BSA (Sigma-Aldrich), then incubated for 90 min with primary antibodies, then a further 30 min with secondary antibodies and DAPI (1:6000 dilution); all antibodies were diluted in PBS + 3% BSA, and are listed below. Wash steps before and after antibody incubations were performed in PBS. All immunofluorescence experiments were imaged as for live cell imaging, but instead using a single camera. Cells were first identified by eye for DAPI signal. Metaphase cells were imaged over 61 z-slices separated by 200 nm (*Figure 1—figure supplement 3*), while interphase cells imaged for chromatic shift calculation were imaged over 121 z-slices separated by 100 nm. Exposure time was 50 ms per z-slice, with laser power in the 488 and 561 nm wavelengths set to: 10% and 30% for images of eGFP-CENP-A / Ndc80-tagRFP; 10% and 10% for eGFP-CENP-A / Nnf1-Alexa594; 10% and 30% for images of eGFP-CENP-A / mCherry-Ndc80; 10% and 30% for CENP-A-Alexa488 / Ndc80-tagRFP; and 20% and 20% for CENP-C-Alexa488 / mCherry-Ndc80, respectively, to optimize imaging. Spot centres were located (*Figure 1—figure supplement 3*), as described below. Sister kinetochores

were manually paired for experiments expressing eGFP-CENP-A and Nnf1-Alexa488 in order to allow calculation of $\Delta_{1D}$. Laser power in the 640 nm wavelength was set to: 10% for images of eGFP-CENP-A / Nnf1-Alexa594; and 40% for images of eGFP-CENP-A / mCherry-Ndc80, for imaging of αTubulin-Alexa647.

## Primary and secondary antibodies

Primary antibodies used were: mouse anti-CENP-A (1:500; Abcam, UK), guinea pig anti-CENP-C (1:2000; MBL, Woburn, MA), rabbit anti-Nnf1 (1:1000; *McAinsh et al., 2006*), and mouse anti-α Tubulin (1:1000; Sigma-Aldrich). Secondary antibodies used were: AlexaFluor-488nm goat anti-mouse (1:500; Invitrogen, Carlsbad, CA), AlexaFluor-488nm goat anti-guinea pig (1:400; Life technologies, Carlsbad, CA), AlexaFluor-594nm goat anti-rabbit (1:500; Invitrogen), and AlexaFluor-647nm goat anti-mouse (1:500; Invitrogen).

## Image processing and analysis

All microscope images and movies were exported from Volocity 6.0 in. OME.TIFF format (The Open Microscopy Environment, UK). They were deconvolved in the 488 and 561 nm wavelengths within Huygens 4.1 (SVI), using point spread functions calculated from 100 nm TetraSpeck fluorescent microspheres (Invitrogen) using the Huygens 4.1 PSF distiller. Images were deconvolved in the 640 nm wavelength within Huygens 4.1 using a theoretical PSF. Deconvolved images were exported from Huygens 4.1 in. r3d format (Applied Precision, Slovakia), and read into MATLAB (R2012a; Mathworks, Natick, MA) using the loci-tools java library (The Open Microscopy Environment). Chromatic shift was calculated as described below. For dynamic live movies, sister kinetochores were detected using the eGFP signal, aligned, tracked and paired in MATLAB as previously described (*Burroughs et al., 2015*). tagRFP fluorescence signal was detected within a hemispherical mask with 300 nm radius directed normal to the metaphase plate and away from the sister pair centre; the hemisphere was centred at the eGFP coordinate, corrected for chromatic shift from the 488 to 561 nm wavelength. 3D Gaussians were then fit to tagRFP spots to find sub-pixel spot centre coordinates (in the same way as is done for eGFP signal), and corrected for chromatic shift into the eGFP wavelength frame of reference. For static live images, the centres of green fluorescence spots were located by 3D Gaussian mixture model fitting (MMF), and red fluorescence signal located, also by Gaussian MMF, within a 300 nm-radius sphere centred at the eGFP spot. Red spot coordinates were then corrected for chromatic shift. Sister kinetochores were manually paired for experiments expressing GFP-CENP-C and Ndc80-tagRFP in order to allow calculation of $\Delta_{1D}$. Custom written software for spot detection and analysis of data is available at http://mechanochemistry.org/mcainsh/software.php. Kinetochore spot coordinate data produced for this study are available at http://figshare.com, DOI: 10.6084/m9.figshare.3859503.

## Delta (Δ) measurement

$\Delta_{3D}$ was measured as the 3D Euclidean distance between eGFP and tagRFP spot centres. All 3D data for untreated and drug-treated cells were filtered so that the *z*-component of $\Delta_{3D}$ (distance between eGFP and tagRFP signal from the same kinetochore) was no larger than 100 nm (half the *z*-pixel resolution). $\Delta_{1D}$ measurements were calculated on this data as before (*Wan et al., 2009*; *Maresca and Salmon, 2009*) using 2D measurements of inter-kinetochore distances. All statistical tests were Mann-Whitney U tests.

## Swivel ($\vartheta$) measurement

$\vartheta$ was calculated using the dot product of the sister-sister and the intra-kinetochore axes. *y*-directional swivel ($\vartheta_y$) of a given kinetochore was calculated using the coordinates of the kinetochore's tagRFP centre and the eGFP centres of each in the kinetochore pair in the *xy*-plane (*Figure 3—figure supplement 1A*). These coordinates form a triangle with sides of length inter-kinetochore distance $d_y$, intra-kinetochore distance $\Delta_{y,1}$, and $\varepsilon_{y,1}$, defined as the distance between a kinetochore's outer marker and its sister's inner marker (*Figure 3—figure supplement 1B*). The cosine rule then allows calculation of *y*-swivel,

$$\vartheta_{y,1} = 180° - \cos^{-1}\left(\frac{d_y^2 + \Delta_{y,1}^2 - \varepsilon_{y,1}^2}{2d_y\Delta_{y,1}}\right)$$

All statistical tests comparing swivel between conditions were Mann-Whitney U tests, except for in the bottom row of *Figure 3B*, where the F-test was used to compare distribution variances of y- and z-directional $\vartheta_{\text{swivel}}$. The kinetochore-microtubule (kMT) axis was manually calculated in the microscope x'y'-axis by drawing a straight line along the length of the αTubulin-Alexa647 signal extending from a kinetochore. Each kinetochore in question was represented solely using its green fluorescence spot centre (i.e. without the red spot visible) to avoid bias derived from the knowledge of the intra-kinetochore axis. $\vartheta_{\text{kMT}}$ was calculated in the y-axis by the same method as y-directional $\vartheta_{\text{swivel}}$ (*Figure 3—figure supplement 1A and B*). The statistical significance of the correlation between the absolute values of $\vartheta_{\text{y-swivel}}$ and $\vartheta_{\text{kMT}}$ on the range [0°, 90°] in Figure 4—figure supplement 2 was calculated using Monte Carlo computation, with $10^6$ permutations; no cases were as extreme as that observed indicating a p-value $< 10^{-6}$. Assuming the correlations (*Figure 4—figure supplement 1D*) conform to a Gaussian distribution, we obtain the estimate p = $2.8 \times 10^{-19}$ (z = 8.9). Correlation was also significant over the full range [–180°, 180°].

## $\Delta_{\text{1D}}$ in terms of $\Delta_{\text{3D}}$ and $\vartheta$

The distance $\Delta_{1D} = \frac{1}{2}\left(d_{y,2} - d_{y,1}\right)$ (see *Figure 2D*) has a complex relationship with the intra-sister distances $\Delta_{y,1}$, $\Delta_{y,2}$ and kinetochore swivels. The setup is shown in *Figure 3—figure supplement 1C*; this is a projection to 2D from 3D. $\vartheta_{y,1}$, $\vartheta_{y,2}$ are the y-swivel of each kinetochore, and $\Delta_{y,1}$, $\Delta_{y,2}$, $d_{y,1}$, $d_{y,2}$ are 2D intra-kinetochore and inter-fluorescent marker projected distances, *i.e.*

$$\Delta_{y,1} = \Delta_{3D,1}\cos\theta_{z,1}$$

etc., in terms of the corresponding z-swivel. Mapping the distances in *Figure 3—figure supplement 1C* onto the sister-sister axis and its perpendicular, we obtain the configuration in *Figure 3—figure supplement 1D*. Pythagoras' theorem then gives

$$d_{y,2}^2 = \left(d_{y,1} + \Delta_{y,1}\cos\vartheta_{y,1} + \Delta_{y,2}\cos\vartheta_{y,2}\right)^2 + \left(\Delta_{y,1}\sin\vartheta_{y,1} + \Delta_{y,2}\sin\vartheta_{y,2}\right)^2$$

This can be rearranged to give,

$$\begin{aligned}\Delta_{1D} &= \tfrac{1}{2}\left(d_{y,2} - d_{y,1}\right)\\ &= \frac{d_{y,1}}{d_{y,1}+d_{y,2}}\left(\Delta_{y,1}\cos\vartheta_{y,1} + \Delta_{y,2}\cos\vartheta_{y,2}\right) + \frac{\left(\Delta_{y,1}^2 + \Delta_{y,2}^2 + 2\Delta_{y,1}\Delta_{y,2}\cos\left(\vartheta_{y,1}-\vartheta_{y,2}\right)\right)}{2\left(d_{y,1}+d_{y,2}\right)}\end{aligned}$$

The distance $\Delta_{1D}$ is thus, to first order in $\Delta_{y,1}/\bar{d}$, $\Delta_{y,2}/\bar{d}$ ($\bar{d}$ the mean of $d_{y,1}$, $d_{y,2}$), a projection onto the sister-sister axis with the approximation, $\Delta_{1D} = \frac{1}{2}\left(\Delta_{y,1}\cos\vartheta_{y,1} + \Delta_{y,2}\cos\vartheta_{y,2}\right) + O\left(\Delta_{y,1}/\bar{d}, \Delta_{y,1}/\bar{d}\right)$, although the second term is positive so $\Delta_{1D}$ will be an underestimate of the 1D projection distance. Since $\Delta_1/\bar{d}$ is of order 0.1 in HeLa cells, this approximation is at best incurring a 10% error. By estimating the swivel from the tilt of the fluorescent spots, Wan *et al* suggest that the projection from 2D to 1D can be corrected, estimating $\frac{1}{2}\left(\Delta_{y,1} + \Delta_{y,2}\right)$ from $\Delta_{1D}$. However, this is challenging even when using this approximation, only being straight forward if $\Delta_{y,1} = \Delta_{y,2}$, which is not the case as these are themselves projections from 3D to 2D. Thus, $\Delta_{1D}$, although attractive as a measurement because it isn't affected by chromatic aberration, is a poor estimate of the intra-kinetochore distance because it is subject to multiple projection effects, and in fact underestimates the true 1D projection distance onto the sister-sister axis (at least to second order).

## Chromatic-dual camera shift correction

Chromatic shift was calculated for each imaging session using 3D image stacks of interphase HeLa-K cells with both green and red fluorescence signal located at the N-terminus of CENP-A. Chromatic shift correction of untreated and taxol-treated live dynamic movies of cells expressing eGFP-CENP-A and Ndc80-tagRFP were calculated using cells expressing both eGFP-CENP-A and mCherry-CENP-A, whilst chromatic shift correction of nocodazole-treated live dynamic movies, all static live and all immunofluorescence images were calculated using cells expressing HaloTag-CENP-A. HaloTag-

CENP-A cells were prepared by simultaneously incubating with 2.5 µM Oregon Green (Promega) and 1 µM TMR (Promega) fluorescent ligands for 15 min, followed by 30 min incubation in ligand-free growth medium before imaging. Green and red fluorescence was imaged simultaneously on two separate CCD cameras with a voxel size of 69.4 × 69.4 × 100 nm. The centres of green spots were located by 3D Gaussian MMF, and red signal located, also by Gaussian MMF, within a 300 nm-radius sphere centred at the green spot. Chromatic shift from green to red signal, $\zeta = [\zeta_x, \zeta_y, \zeta_z]$, also incorporating mis-alignment of the two cameras, was calculated as the 3D vector pointing from the green to the red spot centres in the microscope coordinate system. Quality of spots was optimised by accepting only green spots with intensity > 25% that of the maximum spot intensity, and those not located within 750 nm of another green spot. Due to a linear dependence in the dual camera setup of $\zeta_z$ on the focal distance away from the glass slide, $\zeta_z$ was shifted by +78.2 nm to adjust for the increased distance of the 3 µm thick region of the metaphase plate imaged in the eGFP/tagRFP channels from interphase to mitotic cells (this shift was not required when using a single camera for imaging, i.e. for the immunofluorescence experiments). The standard deviation of $\zeta$ distributions, averaged over all live dynamic experiments, were 25.5, 23.1 and 43.0 nm for $\zeta_x$, $\zeta_y$ and $\zeta_z$ respectively. To validate our calculated chromatic shift we analysed the distribution of $\Delta$ in the *x*, *y* and *z* direction for the live dynamic untreated movies (*Figure 1—figure supplement 4B*). $\Delta_x$ should be bimodal and symmetrical about zero, because the Ndc80 and CENP-A signals are on average orientated along the normal to the metaphase plate pointing at one of the two spindle poles. The $\Delta_x$ distribution was bimodal with median value 5.4 (± 1.2) nm (n = 4291; *Figure 1—figure supplement 4C*). We expect $\Delta_y$ and $\Delta_z$ to be centred around zero because on average Ndc80 and CENP-A should not be displaced relative to each other in the *y* or *z* direction. This is the case with the median $\Delta_y$ and $\Delta_z$ values being 8.8 (± 1.0) nm and –2.9 (± 0.9) nm respectively (each n = 4291; *Figure 1—figure supplement 4C*). Thus, our shift correction from interphase cells when applied to metaphase cells recreates the known average geometry for the kinetochore inner and outer plates.

## Acknowledgements

We thank Emanuele Roscioli, Masanori Mishima and Rob Cross for helpful discussions regarding the manuscript. We are also grateful to E Nigg for Ndc80 cDNA, V Silio for Bub3-eGFP, P Meraldi for GFP-CENP-O cells, A Musacchio for GFP-CENP-C cells and T Hirota for eGFP-CENP-A / mCherry-Mis12 cells.

## Additional information

### Funding

| Funder | Grant reference number | Author |
| --- | --- | --- |
| Engineering and Physical Sciences Research Council | EP/F500378/1 | Chris A Smith |
| Wellcome Trust | 106151/Z/14/Z | Andrew D McAinsh |
| Biotechnology and Biological Sciences Research Council | BB/I021353/1 | Andrew D McAinsh Nigel J Burroughs |

The funders had no role in study design, data collection and interpretation, or the decision to submit the work for publication.

### Author contributions

CAS, Conception and design, Acquisition of data, Analysis and interpretation of data, Drafting or revising the article; ADM, NJB, Conception and design, Analysis and interpretation of data, Drafting or revising the article

### Author ORCIDs

Chris A Smith, http://orcid.org/0000-0002-5035-6828
Andrew D McAinsh, http://orcid.org/0000-0001-6808-0711
Nigel J Burroughs, http://orcid.org/0000-0002-4632-1550

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
