## [Decision Letter]

Thank you for submitting your article "Human kinetochores are swivel joints that mediate microtubule attachments" for consideration by *eLife*. Your article has been favorably evaluated by Vivek Malhotra (Senior Editor) and three reviewers, one of whom is a member of our Board of Reviewing Editors.

The reviewers have discussed the reviews with one another and the Reviewing Editor has drafted this decision to help you prepare a revised submission.

Summary:

Intra-kinetochore distance (δ) measurements have a strong impact on the discussion of the physical changes kinetochores undergo when binding to microtubules, and of their influence on the spindle assembly checkpoint (SAC) and kinetochore-microtubule attachment. Most prior δ measurements have not systematically taken into account the displacement of kinetochores in the z-axis. This work addresses a gap in the field by tracking metaphase kinetochore pairs in 3D in live cells and measuring δ in unperturbed cells and cells treated with microtubule drugs that activate the SAC. In contrast with previous work, this study finds that there is no significant change in 3D distance between the inner and outer kinetochore. This calls into question the role of changes in δ as a signal for SAC satisfaction. The authors suggest that the intrakinetochore distance changes others report in 2D and 1D measurements actually occur because of a displacement of the outer kinetochore relative to the inner kinetochore, a feature they term "swivel". They find that kinetochore swivel increases in response to treatment with nocodazole and decreases as kinetochore-microtubule attachments form and mature. These findings identify a previously undescribed feature of kinetochore structure and challenge the existing model that intrakinetochore distance changes control the SAC. Swivel reflects the intrinsic mechanical flexibility of kinetochore structure, which may allow the kinetochore to stably maintain attachment to stiff microtubules.

The quality of the data shown in this manuscript is generally high and most conclusions are supported by experiments. The characterization of swivel as a new degree of freedom in kinetochore architecture is convincing and brings new insight into the dynamics of kinetochore structure (or lack thereof) during mitotic spindle assembly. The suggestion that swivel has been so far mistaken for actual changes in intra-kinetochore stretch, causing the distance between the inner and outer plates to be sometimes underestimated when measured in 1D, is also compelling.

The study's main conclusions, however, need further support by additional experiments and data analysis addressing the discrepancies between the presented results and previously reported changes in 2D and 3D intra-kinetochore stretch. If the source of discrepancy could be identified, the manuscript would certainly have a durable impact in the field.

Essential revisions:

The main conclusion of the manuscript is that the particular intra-kinetochore linkages monitored by the authors are not force compliant. This is an interesting and important conclusion, and the approach used here sets a standard for the field. The conclusions of the study are limited to the particular distance measurements chosen by the authors. Ultimately, the authors should attempt to address the discrepancy between this paper's results and the results of Etemad et al. Nat Comm 2015. Etemad et al. used CENP-C and the N-terminal region of Hec1 to calculate 3D intrakinetochore distances in fixed cells and found that the distance changes significantly in nocodazole. This suggests that the present manuscript's findings may reflect a different probe choice (e.g. CENP-A vs. CENP-C or N- or C-termini of Ndc80/Hec1). Alternatively, the difference might arise because Etemad et al. fixed cells where the authors use live cells. In either case, it would be important to identify the source of the discrepancy.

In particular:

1) CENP-A overexpression may lead to broad CENP-A localization and create complications with centroid determination and its interpretation (e.g. see Discussion in Wan et al. Cell 2009). For these reasons, the authors may consider using CENP-C as another control – and provide population-level quantitative findings for it.

2) Addition of data with a more external outer kinetochore marker would have allowed to address the important question whether the flexibility of the Ndc80 complex is enabling force-compliance. One would also like to see δ measurements addressing the outer kinetochore (like in the case of Etemad et al. 2015), e.g. RFP-Nuf2 to GFP-CENP-A (we are aware that genetically encoded probes at the N-terminus of Hec1/Ndc80 leads to chromosome alignment errors).

3) If no change is observed with the new probes, it would be important to clarify if the source of the difference is created by the fixation.

4) The authors make a potentially nice point that the swivel angle and k-fiber angle match (Figure 4 and Figure 4—figure supplement 1), but do not show any population-level statistics/quantification of this co-linearity. This must be provided as conclusions cannot be made from single examples without knowing whether they represent population-level behaviour or not.

---

## [Author Response]

*The study's main conclusions, however, need further support by additional experiments and data analysis addressing the discrepancies between the presented results and previously reported changes in 2D and 3D intra-kinetochore stretch. If the source of discrepancy could be identified, the manuscript would certainly have a durable impact in the field.*

Essential revisions:

*The main conclusion of the manuscript is that the particular intra-kinetochore linkages monitored by the authors are not force compliant. This is an interesting and important conclusion, and the approach used here sets a standard for the field. The conclusions of the study are limited to the particular distance measurements chosen by the authors. Ultimately, the authors should attempt to address the discrepancy between this paper's results and the results of Etemad et al. Nat Comm 2015. Etemad et al. used CENP-C and the N-terminal region of Hec1 to calculate 3D intrakinetochore distances in fixed cells and found that the distance changes significantly in nocodazole. This suggests that the present manuscript's findings may reflect a different probe choice (e.g. CENP-A vs. CENP-C or N- or C-termini of Ndc80/Hec1). Alternatively, the difference might arise because Etemad et al. fixed cells where the authors use live cells. In either case, it would be important to identify the source of the discrepancy.*

We have repeated the experiments of Etemad et al., Nat Comm, 2015 measuring the intra-kinetochore distance from CENP-C (using anti-CENP-C antibodies) to the amino-terminus of Ndc80 (mCherry-Ndc80) in paraformaldehyde fixed cells (Figure 1—figure supplement 2). We found smaller changes of 3D δ under nocodazole treatment compared to Etemad (17 nm vs. 25 nm). This may reflect fixation effects (see Magidson et al., JCB, 2016) and/or that Etemad et al. used the centre of mass to identify spot positions based on 147 measurements. In our experiment we used a more accurate Gaussian mixture model (>8000 measurements) and also did not arrest cells in MG132. We also note that in live cells the distance from GFP-CENP-C to mCherry-Ndc80 reduces by only 10 nm in nocodazole and therefore fixation may inflate this particular measurement.

*In particular:*

*1) CENP-A overexpression may lead to broad CENP-A localization and create complications with centroid determination and its interpretation (e.g. see Discussion in Wan et al. Cell 2009). For these reasons, the authors may consider using CENP-C as another control – and provide population-level quantitative findings for it.*

We have now added data for distance between GFP-CENP-C and the carboxy (C)-terminus of the Ndc80 complex (Ndc80-tagRFP), and also performed fixed cell experiments with anti-CENP-A using plain HeLa-K cells (Figure 1—figure supplement 2). These data support the original eGFP-CENP-A data.

2) Addition of data with a more external outer kinetochore marker would have allowed to address the important question whether the flexibility of the Ndc80 complex is enabling force-compliance. One would also like to see δ measurements addressing the outer kinetochore (like in the case of Etemad et al. 2015), e.g. RFP-Nuf2 to GFP-CENP-A (we are aware that genetically encoded probes at the N-terminus of Hec1/Ndc80 leads to chromosome alignment errors).

We have added measurements for the N-terminus of Ndc80 (mCherry-Ndc80), which lies further out than its C-terminus. We find a 10 nm reduction in 3D δ following 2 hr nocodazole treatment that decomposes as a 5 nm reduction in the inner domain, and 5 nm in the outer. However, these changes in 3D distance are smaller than the effect of swivel in changing the proximity between the inner and outer domains, as evidenced by the substantially larger changes in the projected distance δ 1D. We also confirmed our original results using two subunits of the Mis12 complex (mCherry-Mis12 and anti-Nnf1 antibodies).

*3) If no change is observed with the new probes, it would be important to clarify if the source of the difference is created by the fixation.*

See above point 1.

*4) The authors make a potentially nice point that the swivel angle and k-fiber angle match (Figure 4 and Figure 4—figure supplement 1), but do not show any population-level statistics/quantification of this co-linearity. This must be provided as conclusions cannot be made from single examples without knowing whether they represent population-level behaviour or not.*

We now quantify the kinetochore-microtubule (kMT)- angle tended with the sister-sister axis (ϑ kMT), and its correlation with kinetochore swivel (ϑ y-swivel). We find that they are significantly correlated (Figure 4—figure supplement 1).